# Case Report of a Neonate with Severe Perinatal Asphyxia: A Multidisciplinary Approach Involving Therapeutic Hypothermia and Physiotherapy

**DOI:** 10.3390/pediatric17040086

**Published:** 2025-08-11

**Authors:** Marcelina Powązka, Maciej Grzeszczuk, Tatiana Jagodzińska, Ewa Syweńki, Rita Suchanska, Ewa Gieysztor

**Affiliations:** 1Scientific Club No. 15 Progressio Infantis, Faculty of Physiotherapy, Wroclaw Medical University, 50-367 Wroclaw, Poland; 2Division of Histology and Embryology, Department of Human Morphology and Embryology, Wroclaw Medical University, 50-368 Wroclaw, Poland; maciej.grzeszczuk@student.umw.edu.pl; 3Neonatal Unit, J. Gromkowski Provincial Specialized Hospital, Koszarowa St. 5, 51-149 Wroclaw, Poland; tatiana.jagodzinska@wp.pl (T.J.); daria78.rs@gmail.com (R.S.); 4Department of Anesthesiology and Intensive Care for Newborns and Children, J. Gromkowski Provincial Specialized Hospital, Koszarowa St. 5, 51-149 Wroclaw, Poland; ewa.sywenki@pwr.edu.pl; 5Faculty of Medicine, Wroclaw University of Science and Technology, Wybrzeże Stanisława Wyspiańskiego 27, 50-370 Wroclaw, Poland; 6Faculty of Physiotherapy, Wroclaw Medical University, 50-367 Wroclaw, Poland

**Keywords:** hypoxic–ischaemic encephalopathy, therapeutic hypothermia, neurodevelopment, neonatal resuscitation, early rehabilitation

## Abstract

Hypoxic–ischaemic encephalopathy (HIE), a leading cause of perinatal mortality and neurological impairment, affects 1–8/1000 live births in developed countries. Therapeutic hypothermia (TH), the standard treatment for moderate to severe HIE, reduces brain injury by lowering metabolic demand and inhibiting apoptosis. This case study presents a full-term female newborn delivered via caesarean section due to intrauterine asphyxia, with meconium aspiration syndrome and severe HIE (Apgar 0/0/0/2). Notwithstanding the presence of multiorgan failure and grade II intraventricular haemorrhage, TH was initiated within six hours. The patient received circulatory and respiratory support, sedation, and nitric oxide. Early rehabilitation was initiated immediately. Neurofunctional assessment using the TIMP test revealed initial delays (16–25th percentile) at 11 weeks of age; however, the subsequent two evaluations, conducted approximately every two weeks, indicated that the patient was within normal developmental ranges. A similar outcome was observed in the AIMS assessment conducted at seven months of age, which also yielded normal results. Despite MRI findings post-TH showing hypoxic and haemorrhagic lesions, the patient achieved normal development. This case demonstrates the effectiveness of combining TH with early physiotherapy in mitigating severe consequences of HIE, such as cerebral palsy and epilepsy. Long-term follow-up remains crucial for detecting later deficits, particularly during school age. The outcome of this case underscores the significance of timely intervention and multidisciplinary care. While TH and rehabilitation have been shown to improve prognosis, ongoing monitoring is crucial to ensure optimal neurological development trajectories.

## 1. Introduction

The issue of hypoxic–ischaemic encephalopathy (HIE) is a significant medical concern, with a mortality rate of approximately 0.98 per 1000 live births [1]. HIE, also known as perinatal asphyxia (PA), is a major cause of perinatal mortality in full-term infants [2]. PA is defined as a syndrome of symptoms resulting from insufficient or impaired oxygen availability (hypoxia) to the brain tissue of the foetus and newborn during the prenatal, intrapartum, or immediate postpartum periods [3]. The incidence of HIE in the perinatal period ranges from 1 to 8 per 1000 live births in highly developed countries, while in developing countries, it reaches nearly 25 per 1000 newborns [4].

PA may lead to permanent brain damage and severe neurological disorders, affecting cognitive, motor, social, and emotional development. HIE is a significant cause of cerebral palsy and epilepsy [1]. It is a severe clinical condition that poses a considerable risk to newborns, especially those born with low birth weight. Infants with low birth weight are more susceptible to the condition, which increases the risk of severe complications [5].

The severity of perinatal hypoxia determines the severity of HIE, which can range from subclinical forms to moderate neurological disorders and severe central nervous system damage [6]. The severity and prognosis of HIE are classified based on the Sarnat scale, which is presented in Table 1 [7].

The mortality rate for infants with mild HIE is approximately 10%, while the incidence of neurodevelopmental disorders reaches 30%. In severe cases, the mortality rate rises to 60%, and most surviving children develop severe disabilities [8].

TH is a treatment method that helps mitigate the complications of brain hypoxia and effectively reduces mortality, involving the cooling of the newborn’s brain and body to a temperature of approximately 33–34 °C, with temperature typically measured rectally [9]. It should be initiated within the first 6 h after birth and maintained for 72 h, followed by gradual rewarming of the newborn [10]. The neuroprotective effects of TH are attributable to its capacity to reduce cerebral metabolism by approximately 5% for each degree Celsius. Furthermore, the treatment has been shown to delay the depolarisation of hypoxic cells, thereby reducing the extent of brain damage, a critical component in the management of certain neurological disorders [11].

In 2010, the American Academy of Paediatrics Neonatal Resuscitation Program and the International Liaison Committee on Resuscitation (ILCOR) included therapeutic hypothermia as the standard treatment for newborns with moderate and severe HIE [12]. The procedure is performed using one of two cooling methods: selective head cooling with mild whole-body hypothermia (selective head cooling—SHC) or whole-body cooling (WBC) [13].

Whole-body cooling (WBC) involves cooling the entire body to a temperature of 33–34 °C using a cooling blanket, with the temperature of the circulating liquid being regulated by a thermostat. In selective head cooling, the recommended temperature is 34.5–35 °C for the head, achieved using the Olympic Cool-Cap System (OCCS), which allows for controlled cooling of the newborn’s head in cases of mild systemic hypothermia while keeping the rest of the body warm [14,15].

In order to initiate head or whole-body cooling in a newborn with HIE, specific criteria must be met. Firstly, the gestational age of the infant must be at least 35 weeks or more. Secondly, the time elapsed since birth must not exceed 6 h, as initiating cooling later may not provide therapeutic benefits [16].

In order to qualify a newborn for therapeutic hypothermia, biochemical criteria (Group A) must be met. Firstly, the Apgar score at 1, 3, 5, and 10 min of life should be 5 or lower. Additionally, if the newborn still requires resuscitation using an endotracheal tube or mask at 10 min of life, this condition is also met. Furthermore, the presence of acidosis within the first hour of life, characterised by an umbilical cord, arterial, or capillary blood pH of 7.0 or lower, is a crucial criterion. Finally, the presence of a base deficit of 16 mmol/L or higher, as detected in umbilical cord, arterial, venous, or capillary blood within the first 60 min of life, serves as an essential biochemical requirement.

Newborns who meet these criteria are then assessed neurologically to determine eligibility for therapeutic hypothermia (Group B). Neurological criteria include the presence of seizures or moderate to severe encephalopathy, manifested by altered consciousness, such as reduced or absent responsiveness to stimuli. Another indicator is abnormal muscle tone, either focal or generalised hypotonia, along with abnormal reflexes, such as a weak or absent sucking reflex and Moro reflex. Autonomic functions are also evaluated, including pupillary response, respiratory pattern, and heart rate. In many centres, aEEG/EEG recordings are considered an additional criterion to aid in the decision-making process for therapy qualification [17]. These criteria are summarised in Table 2.

## 2. Case Presentation

### 2.1. Medical Management

This retrospective case study analysed the therapeutic effects in a female newborn of West Slavic ethnicity, born at 39 weeks of gestation via caesarean section due to imminent intrauterine asphyxia. The patient exhibited symptoms of severe perinatal asphyxia with meconium aspiration. Intensive resuscitation measures were undertaken, and therapeutic hypothermia was initiated (Figure 1). The pregnancy was largely uncomplicated, except for maternal anaemia, which was treated pharmacologically. Three hours prior to delivery, green amniotic fluid was observed, indicating the presence of meconium in the amniotic fluid. The delivery was performed via cesarean section due to fetal distress, and the newborn was in a state of severe perinatal asphyxia.

The Apgar scores in the first minutes were 0/0/0/2. Following resuscitation efforts and intubation, epinephrine was administered, resulting in the restoration of a heartbeat at 10 min of life. The patient was deemed eligible for therapeutic hypothermia and was admitted to the XVII Department of Neonatal and Pediatric Anesthesiology and Intensive Care at J. Gromkowski Regional Specialist Hospital in Wrocław in a critical condition. Sedation, empirical antibiotic therapy, and parenteral nutrition were initiated immediately. A series of diagnostic procedures were conducted, encompassing laboratory analyses of blood (Table A1) and urine samples (Table A2), transcranial ultrasound imaging, and additional investigative modalities. The newborn required haemodynamic support, receiving a dobutamine infusion at a dose of 10 μg/kg/min, which was discontinued on the fifth day of treatment. Due to signs of shock (hypotension 45/17 mmHg (MAP 26), 63/21 mmHg (35), 66/29 (41) mmHg; tachycardia 162/min, 168/min) on the first day, norepinephrine was also administered.

In response to physical stimulation, the patient demonstrated movement of all four limbs and dilation of the pupils. The skin was noted to be pink and elastic, with increased and symmetrical muscle tone. Auscultation revealed intensified breath sounds, suggesting involvement of the respiratory system. The abdomen was described as soft and level, with no evidence of peristalsis.

The aEEG recording 10 min after admission to the neonatal intensive care unit, before HT, showed a broad, moderately abnormal pattern. A cranial ultrasound revealed signs of brain oedema and asymmetry of the choroid plexuses, suggesting intraventricular haemorrhage (IVH). However, IVH was subsequently not confirmed by magnetic resonance imaging (MRI) on day 26 of life. To stabilise coagulation parameters, fresh frozen plasma was transfused. From the outset, the newborn was in an unstable condition requiring respiratory and circulatory support, and multiorgan failure was diagnosed, affecting liver, kidney, respiratory, coagulation, and circulatory functions. Transfontanellar ultrasound (day 1 of life) revealed asymmetry of the lateral ventricles, with the occipital horn measuring 13.4 mm on the right and 20.7 mm on the left. The brain parenchyma showed blurred differentiation and increased echogenicity in the frontal and parietal lobes, while sulcal and gyral patterns remained normal. The corpus callosum was present and measured 3 mm in height. No displacement of the median fissure was observed. The cerebellum and posterior fossa structures appeared normal. Extracerebral fluid spaces were within normal limits bilaterally. Abdominal ultrasound (day 1 of life) showed a normally sized liver (66 × 44 mm) with homogeneous echotexture, no biliary dilation or focal changes, and a thin-walled gallbladder without stones. The kidneys were of normal size and echogenicity bilaterally, without hydronephrosis or collecting system dilatation. The urinary bladder was empty, and the adrenal glands appeared normal.

Due to meconium aspiration, MAS (Meconium Aspiration Syndrome) developed, leading to severe pulmonary hypertension, which was treated with nitric oxide (NO) from day 4 to day 9 of life. Chest radiographs performed on days 1 and 4 (Figure A1) of life revealed ground-glass opacities in both lung fields, consistent with severe Respiratory Distress Syndrome (RDS), with a noticeable progression of changes on the follow-up examination (Table A3). From admission, mechanical ventilation requiring high pressures (PEEP 8 cm H_2_O, PS 18 cm H_2_O) and 100% oxygen was used. After stabilisation of circulatory and respiratory conditions and a reduction in pulmonary hypertension symptoms, the patient was extubated on day 12 of hospitalisation. Due to the patient’s ongoing respiratory distress, non-invasive ventilation was continued until day 20 of life.

Following the initial day of oliguria, high-dose furosemide was administered to induce diuresis, which was successful. Despite the achievement of diuresis, biochemical markers of renal dysfunction—an increase in urea and creatinine levels—continued to increase. Thereafter, kidney failure progressed to the polyuric phase, necessitating treatment with Minirin (active ingredient: desmopressin) from day 7 to day 14 of life.

Neurologically, despite adverse perinatal circumstances, there was no suspicion of brain death. Cranial ultrasound revealed a previous grade II intracranial haemorrhage on the left side of the cranium. It was not possible to perform a head MRI during the first week of life due to the patient’s critical condition. Upon awakening, which occurred on day 11, symmetrical but variably increased muscle tone was observed, and the sucking and rooting reflexes were absent. However, the patient did not require oral suctioning, was not at risk of aspiration, and demonstrated stable independent breathing for several days, indicating no difficulty swallowing saliva or clearing secretions. A neurological assessment revealed no Galant reflex.

On day 26 of life, the infant was transferred in a stable condition to the XVII Neonatology Department of the same hospital. During her stay, her general condition remained good, with continuous monitoring of vital parameters. The infant was noted to be in a stable respiratory and circulatory state, with no observed seizures. Neurological consultations were conducted, and a cranial ultrasound showed no significant abnormalities. MRI revealed a single hyperintense lesion near the posterior part of the right lateral ventricle, measuring 3 mm in diameter, without diffusion restriction, indicating a hypoxic–ischaemic lesion and haemorrhagic changes. An ophthalmologic examination showed no abnormalities. The infant was fed maternal milk, initially via a feeding tube. Through a series of meticulous physiotherapy interventions, the infant demonstrated a progressive enhancement in sucking proficiency, thereby superseding the necessity for tube feeding. The infant was required to receive intragastric probe feeding until it reached 41 days of age. A systematic monitoring of weight gain was also conducted, which revealed an uninterrupted progression. At the age of seven weeks, the infant, in a stable condition, was discharged from the hospital in the absence of any distressing symptoms or indications. This discharge was facilitated by the assurance of adequate respiratory and circulatory function, as well as the infant’s ability to be fully fed orally. During long-term follow-up, a transfontanellar ultrasound performed at 6.5 months of age demonstrated normal and symmetrical ventricles (K/M index: 32%), thalamo-occipital diameters of 12 mm (right) and 13 mm (left), and no abnormalities in the choroid plexus, corpus callosum, or cerebral tissue.

### 2.2. Physiotherapeutic Management

In the XVII Department of Neonatal and Paediatric Anesthesiology and Intensive Care, the following physiotherapeutic interventions were undertaken for the patient: stimulation of the grasp reflex, passive exercises for the arms and legs, foot and hand massage, chest percussion in drainage positions, sucking stimulation, sensory stimulation of the oral area, and intraoral massage. During tube feeding, sucking reflex stimulation was applied using a bottle nipple secured with gauze, along with taste stimulation using milk drops.

During the patient’s stay at the XVIII Neonatology Department, in addition to the aforementioned activities, therapy based on the NDT-Bobath Concept was conducted. This was aimed at normalising muscle tone and supporting the patient’s adaptive and self-regulation processes. Rehabilitation was then continued on an outpatient basis. The patient was assessed on three occasions at short intervals (approximately two weeks apart) using the TIMP test (Figure 2), an internationally recognised method for evaluating the psychomotor development of infants up to five months of age, known for its high predictive value regarding future developmental disorders. In the initial assessment at 11 weeks of age, the patient scored between the 16th and 25th percentile, indicating normal but slower/delayed development.

However, subsequent assessments yielded results above the 25th percentile, indicating a fully normal outcome. As the test is designed for children up to 5 months of age, subsequent assessments were conducted using the equally reliable AIMS scale (Figure 3). By seven months, the patient’s performance was within normal limits again. Normal motor patterns, muscle tone, manipulative skills and interactions with people and the environment were also observed.

## 3. Discussion

TH is the only treatment currently approved for neonatal HIE, with the capacity to improve outcomes in infants with HIE [18]. The presented case of a newborn born in a state of extreme asphyxia, with an extremely poor initial Apgar score (0/0/0/2) and symptoms of multiorgan failure, represents an exceptionally challenging clinical scenario, where time and medical decisions play a crucial role. Whilst therapeutic hypothermia is a method with a well-established safety profile, there are certain risks associated with its use. Monitoring vital parameters and biochemical markers of hypoxia is crucial for the safety and effectiveness of TH. Furthermore, it is also essential to monitor hypothermia, hyperoxia, hypercapnia, hypoglycaemia, and hypocalcaemia during the first hours of life [19,20]. The degree and duration of hypothermia, the method of rewarming, and the concurrent use of other therapeutic interventions play a decisive role. The administration of TH in newborns necessitates substantial involvement from intensive care units. It is imperative to acknowledge that during TH, the management of concomitant conditions, including but not limited to seizures, brain oedema, respiratory failure, acute kidney injury, and electrolyte or metabolic disorders, is a critical component of the therapeutic regimen [21,22].

Such conditions are often associated with a high risk of severe neurological complications, including cerebral palsy, epilepsy, and significant delays in psychomotor development [23]. A notable aspect of this case is that, despite MRI results from day 26 findings of hypoxic and haemorrhagic lesions, the patient demonstrated normal development in key areas from the third month of life. This suggests that early intervention can significantly modify the natural course of HIE, even in its most severe forms. This outcome is consistent with the observations of Kubisa et al. [24], who demonstrated that children treated with TH during the first year of life exhibited normal psychomotor development. Nevertheless, long-term follow-up remains necessary, particularly with respect to cognitive and social functions in school-age children, as subtle deficits may become apparent later in life.

TH is currently recognised as the standard of care for newborns with moderate and severe HIE and is the only therapeutic approach with proven efficacy in reducing the risk of permanent neurological damage [10]. The mechanism of TH involves slowing neuronal metabolism and limiting apoptotic processes, effectively reducing the extent of brain injury caused by hypoxia. Studies show that early application of TH (within six hours of birth) significantly increases the chances of improved neurological function in newborns [25]. The hypothermia protocol was implemented in accordance with prevailing guidelines, a factor that may have been instrumental in averting severe neurological complications [26]. Another salient factor that may have contributed to the remarkably favourable prognosis was the intensive physiotherapy regimen initiated in the neonatal intensive care unit and subsequently continued in an outpatient setting. Given the risk factors for developmental disorders and muscle tone abnormalities in the newborn, the implementation of an early physiotherapy intervention programme was substantiated [27]. The utilisation of methodologies derived from the NDT-Bobath concept [28,29], in conjunction with the monitoring of progress through the utilisation of TIMP and AIMS tests [30,31], facilitated continuous evaluation and remediation of potential deficiencies. Research findings suggest that the implementation of early rehabilitative interventions in newborns following HIE can result in substantial enhancement of long-term functional capabilities [32]. Initial psychomotor development assessments conducted at 11 weeks of age indicated a delay (16th–25th percentile); however, subsequent evaluations revealed fully normal development, a remarkable outcome in such severe cases. The integration of TH with rehabilitation has been shown to yield substantial benefits in terms of long-term outcomes [33].

Similar rare cases have been described in the literature. Bhandary et al. [34] presented a case series of six neonates with severe HIE who underwent therapeutic hypothermia while being treated with extracorporeal life support (ECLS). Despite the presence of multiorgan failure, fluid overload, and intracranial haemorrhage in some cases, neurodevelopmental follow-up between 6 and 36 months revealed favourable outcomes in the majority of patients. Additionally, Okulu et al. [35] reported a case of a neonate with meconium aspiration syndrome and severe HIE who underwent both ECMO and whole-body hypothermia without any circuit or bleeding complications. The infant achieved age-appropriate psychomotor development at 24 months. These cases support the notion that intensive early intervention, even in the most critical scenarios, can significantly improve long-term outcomes in neonates with HIE. These findings further underscore the importance of individualised, multidisciplinary approaches in the management of severe HIE.

Despite the encouraging results, ongoing monitoring of the child’s development remains paramount, particularly during the preschool and school years. Research has indicated that even children who do not manifest substantial deficits early in life following HIE may subsequently encounter learning difficulties, attention disorders, or coordination impairments [36]. Consequently, the provision of additional support and continuous monitoring of the girl’s development, particularly in the domains of cognition, motor abilities, and social interaction, is imperative [37].

In conclusion, the efficacy of hypothermia and rehabilitation in this case underscores the significance of prompt intervention. However, ongoing monitoring of the child is imperative to facilitate optimal development and provide support in subsequent stages of life. There is a paucity of research evaluating the long-term consequences of TH on cognitive functions, memory, and adaptive abilities in children [38]. Therefore, there is a pressing need for additional randomised controlled trials with adequately powered cohorts, whose results could provide a stronger foundation for refining clinical guidelines on the use of therapeutic hypothermia in neonates with severe hypoxic–ischaemic encephalopathy.

## 4. Conclusions

It is noteworthy that despite the severe initial condition, the girl is currently developing normally due to the continuation of resuscitation efforts in this full-term newborn. The implementation of both therapeutic hypothermia and comprehensive physiotherapy protected her from disability, which is often expected in the context of such severe perinatal complications. This case suggests the effectiveness of therapeutic hypothermia and comprehensive physiotherapy in preventing permanent neurological damage in newborns with severe HIE. The timely administration of both therapeutic modalities by a skilled medical team has resulted in the girl exhibiting the potential for a healthy life. Nevertheless, ongoing monitoring is imperative to ensure optimal development and provide support at subsequent stages of life.

## Figures and Tables

**Figure 1 pediatrrep-17-00086-f001:**
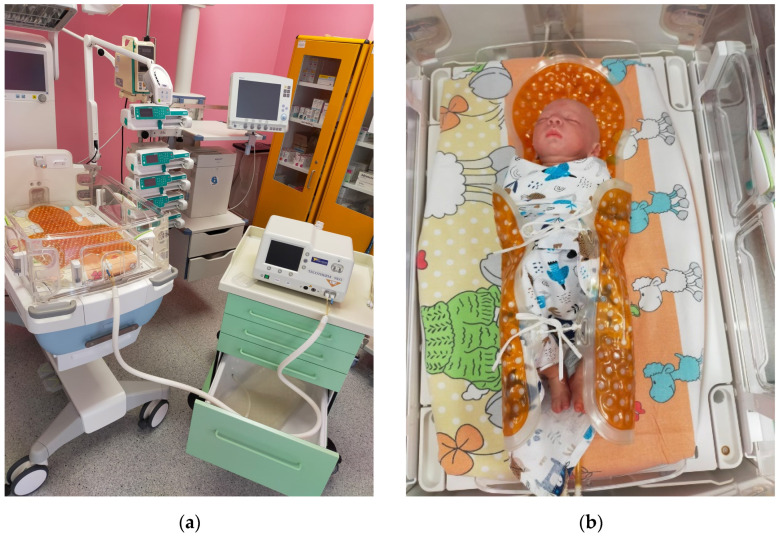
Performance of hypothermia treatment. (**a**) Stand for performance of therapeutic hypothermia treatment, with visible incubator and medical apparatus. (**b**) Doll illustrating the position of a newborn during the treatment, lying in an incubator on a special cooling mattress.

**Figure 2 pediatrrep-17-00086-f002:**
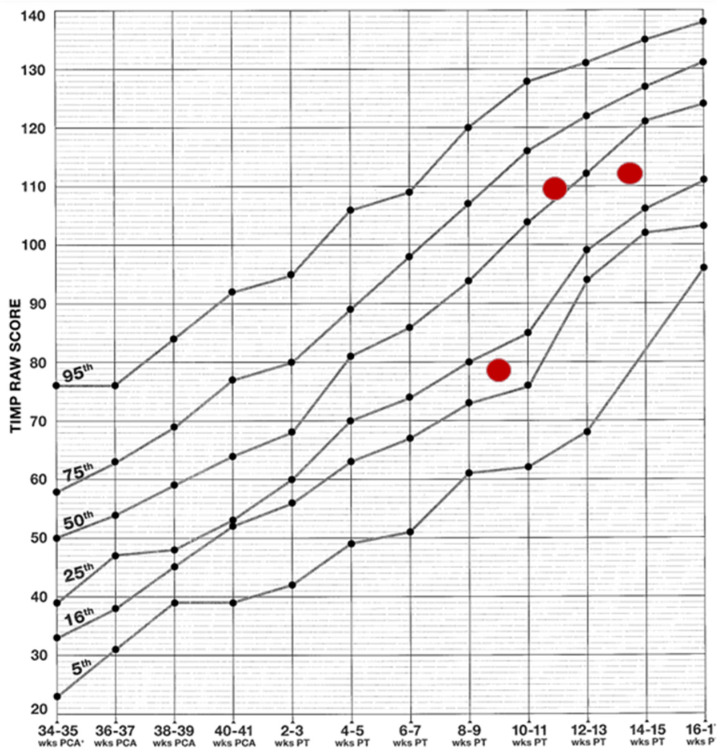
Results achieved on the TIMP scale. The graph presents the raw scores from the Test of Infant Motor Performance (TIMP) in relation to normative percentiles. Red dots represent the patient’s scores from successive assessments, indicating an improvement in motor function over time.

**Figure 3 pediatrrep-17-00086-f003:**
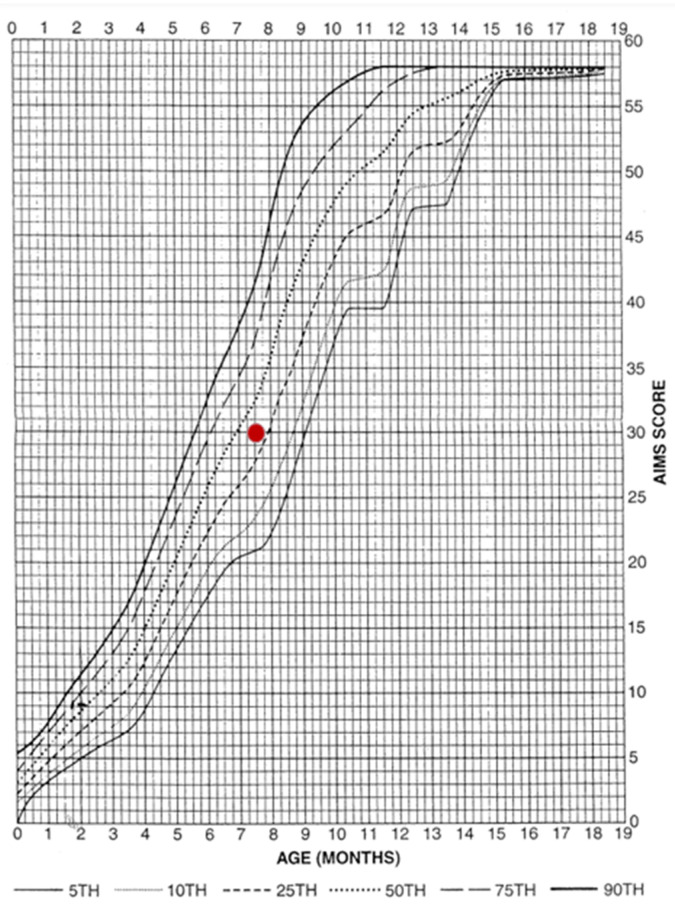
Result achieved on the AIMS scale. The graph presents the patient’s score on the Alberta Infant Motor Scale (AIMS) against normative percentiles. The red dot indicates the obtained score, which falls within the normal range for the child’s age.

**Table 1 pediatrrep-17-00086-t001:** **Sarnat Scale [7]**. HIE (Hypoxic–Ischaemic Encephalopathy).

Symptoms	Mild	Moderate	Severe
Consciousness	Hyperalertness	Lethargy/drowsiness	Coma
Spontaneous activity	Normal	Reduced	Absent
Neuromuscular tone			
Posturing	Moderate distal flexion	Strong distal flexion	Decerebration
Muscle tone	Normal	Hypotonia	Flaccidity
Reflexes			
Sucking reflex	Active	Weak	Absent
Moro reflex	Exaggerated	Weak	Absent
Neck tone	Weak	Strong	Absent
Autonomic Functions			
Pupils	Dilated	Constricted	Variable, poor light reflex
Heart rate	Tachycardia	Bradycardia	Variable
Respiration	Regular	Variable rate	Apnea
Seizures	Absent	Frequent	Rare
Duration of HIE	<24 h	24–72 h	>72 h

**Table 2 pediatrrep-17-00086-t002:** Criteria qualifying for therapeutic hypothermia [17].

Group	Criteria	Details
Group A (Biochemical Criteria)	Low Apgar Score	Apgar score of 5 or lower at 1, 3, 5, and 10 min of life
	Prolonged Resuscitation	Need for resuscitation with endotracheal tube or mask at 10 min of life
	Acidosis	pH ≤ 7.0 in umbilical cord, arterial, or capillary blood within the first hour of life
	Base Deficit	Base deficit ≥ 16 mmol/L in umbilical cord, arterial, venous, or capillary blood within the first 60 min of life
Group B (Neurological Criteria)	Seizures	Presence of seizures
	Moderate to Severe Encephalopathy	Altered consciousness (e.g., reduced or absent responsiveness to stimuli)
	Abnormal Muscle Tone	Focal or generalised hypotonia
	Abnormal Reflexes	Weak or absent sucking reflex and Moro reflex
	Impaired Autonomic Functions	Abnormal pupillary response, respiratory pattern, or heart rate
	EEG/aEEG Findings (optional)	EEG/aEEG recordings may be used as an additional supportive criterion

## Data Availability

The raw data will be made available to other researchers upon reasonable request. To access protocols or datasets, contact ewa.gieysztor@umw.edu.pl.

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
