# Peer review of "Case Report of a Neonate with Severe Perinatal Asphyxia: A Multidisciplinary Approach Involving Therapeutic Hypothermia and Physiotherapy"

_pediatrrep, 2025, doi:10.3390/pediatric17040086_

Round 1
Reviewer 1 Report
Comments and Suggestions for Authors
Dear Authors,
I had the opportunity to review the paper ”Case Report of a Neonate with Severe Perinatal Asphyxia: A Multidisciplinary Approach Involving Therapeutic Hypothermia and Physiotherapy”.
Congratulations, despite some minor suggestions/observations, the manuscript objectively relates to one of the rare cases of severe asphyxia with a good outcome at medium-term follow-up.
The Title is correctly summarizing the content of the manuscript
The Abstract is well-written, providing sufficient information on the content for readers.
The Introduction, in my opinion, is too long; I suggest moving the details on hypoxic-ischemic encephalopathy and therapeutic hypothermia to the Discussion chapter.
The Case Report:
- Please describe the shock signs for which norepinephrine was administered.
- Images of aEEG and brain ultrasound would be relevant and informative, and important to be offered.
- Asymmetry of the choroid plexus is not similar to intraventricular hemorrhage; it could be a normal variant. Additionally, the intraventricular hemorrhage was not confirmed by the follow-up MRI, at least not in the ventricles, as described in the report.
- Lung radiography would also be beneficial to see, along with the values of the blood tests indicating multiorgan failure.
- Could you please provide some references for the Bobath concept and the TIMP test?
The Discussions are objective, primarily focusing on the importance of early and intensive therapeutic hypothermia and physiotherapy, the main message of the report. Yet, discussions on similar, despite rare, cases of neonatal severe hypoxic-ischemic encephalopathy are missing. This would also be of interest.
The conclusions align with the presented case.
Thank you
Author Response
Dear Reviewer,
We would like to thank you for your time and constructive review of our work. We have followed your suggestions and we corrected the manuscript. We believe that these corrections improve our work and make it easier for readers to understand.
The Introduction, in my opinion, is too long; I suggest moving the details on hypoxic-ischemic encephalopathy and therapeutic hypothermia to the Discussion chapter.
Thank you for this suggestion. We have moved some of the information from the Introduction to the Discussion section.
Please describe the shock signs for which norepinephrine was administered.
We have added the clinical signs of shock to the manuscript: hypotension (45/17 mmHg [MAP 26], 63/21 mmHg [MAP 35], 66/29 mmHg [MAP 41]) and tachycardia (162/min, 168/min).
Images of aEEG and brain ultrasound would be relevant and informative, and important to be offered.
In our center, aEEG and ultrasound images are not archived; only their descriptions are recorded. We have added the descriptions of both the cranial and abdominal ultrasound examinations. Unfortunately, we do not have access to the actual aEEG recordings.
Asymmetry of the choroid plexus is not similar to intraventricular hemorrhage; it could be a normal variant. Additionally, the intraventricular hemorrhage was not confirmed by the follow-up MRI, at least not in the ventricles, as described in the report.
At the time of the examination, the choroid plexus asymmetry was suggestive of a potential abnormality, which we believed was worth highlighting. Following your comment, we have added an immediate follow-up sentence clarifying that intraventricular hemorrhage was not confirmed by MRI, in order to avoid misleading the reader.
Lung radiography would also be beneficial to see, along with the values of the blood tests indicating multiorgan failure.
Thank you very much for the suggestion. We have included the blood and urine test results from the first days of the patient’s life as Appendix A, and comparative descriptions of the chest X-rays in table format as Appendix B. We have also added the corresponding X-ray images.
Could you please provide some references for the Bobath concept and the TIMP test?
We have added appropriate references regarding the NDT Bobath method and the TIMP test.
The Discussions are objective, primarily focusing on the importance of early and intensive therapeutic hypothermia and physiotherapy, the main message of the report. Yet, discussions on similar, despite rare, cases of neonatal severe hypoxic-ischemic encephalopathy are missing. This would also be of interest.
We have addressed two rare cases reported in other publications in the Discussion section. This addition enriches the discussion while maintaining conciseness and focus.
Additionally, we have thoroughly reviewed the entire manuscript and made necessary corrections. We have also prepared supplementary materials that will be included as an additional part of the article; however, incorporating them directly into the main text would have disrupted its clarity and flow.
Best regards,
Marcelina Powązka

Reviewer 2 Report
Comments and Suggestions for Authors
First of all, I would like to sincerely congratulate everyone who participated in the treatment of the girl described in the article on the excellent outcome. And since there are not many contributions in the literature describing complementary treatments (including physiotherapy), the article is worth publishing.
However, it should be emphasized that this is a presentation of one case and the results are difficult to evaluate from only one perspective. This is also why I would use the term "suggests" instead of "confirms" in line 296. It would also be necessary to add a sentence to the conclusions about the need for additional RCTs with a sufficient number of children included, the results of which could be the basis for better guidelines.
Some specific comments:
line 51: for better understanding of the problem I suggest that the sentence in lines 51 - 52 would be moved before line 44, as it shows the proportion of HIE as a cause of total perinatal death;
line 64: "Tables may have the footer" probably originates from the basic instructions for preparing the table?;
lines 95 - 111: criteria for HT could be summarised in a table that would be very useful to the reader;
line 109: some additional explanation would be expected as aEEG is applied in most centers that perform HT, as Group C (as I understand from line 159, aEEG was also used in the case of the girl described);
lines 123 - 125: I miss more specific recommendation on timing of MRI; in most centres it is performed between day 5 and 7; according to line 191, in your case it has been performed after day 26? - so it could hardly be characterised as "initial" (line 248);
line 159: "aEEG recording after the initial 10 minutes" - unclear, please specify: after birth? / after admission to NICU? - after initiation of HT?;
line 168: more data on "high pressures" are needed (in cm H2O);
line 173: words in brackets regarding active component furosemide are abundant;
lines 174 - 177: I doubt that furosemide was the cause for high urea / creatinine (I also miss the data on timing of these high values), more likely it was the result of systemic perinatal asphyxia;
line 178: US is not the method to confirm brain death;
line 194: as "vision" is not tested, I would suggest to use expressions "an ophtalmologic examination revealed no abnormalities";
line 197: how long (in days) did the infant need tube feeding?;
The article contains a lot of listing of signs/symptoms/assessments without a "red" thread or structured description (e.g. lines 180 - 185).
A few typo mistakes:
line 51: 0. 98 (correct: 0.98) per 1,000 live births;
line 100: pH of 7. 0 (correct 7.0);
line 215: Rehabilitatio0n.
Authors should also check that the list of references complies with the instructions for authors.
Author Response
Dear Reviewer,
We would like to thank you for your time and constructive review of our work. We have followed your suggestions and we corrected the manuscript. We believe that these corrections improve our work and make it easier for readers to understand.
However, it should be emphasized that this is a presentation of one case and the results are difficult to evaluate from only one perspective. This is also why I would use the term "suggests" instead of "confirms" in line 296. It would also be necessary to add a sentence to the conclusions about the need for additional RCTs with a sufficient number of children included, the results of which could be the basis for better guidelines.
Thank you for this valuable comment. We have addressed it and made the necessary changes to the text.
line 51: for better understanding of the problem I suggest that the sentence in lines 51 - 52 would be moved before line 44, as it shows the proportion of HIE as a cause of total perinatal death;
We have followed your suggestion and reorganized the Introduction accordingly.
line 64: "Tables may have the footer" probably originates from the basic instructions for preparing the table?;
Yes, this was a remnant of the manuscript template. It has now been removed.
lines 95 - 111: criteria for HT could be summarised in a table that would be very useful to the reader;
We agree, and we have summarized all the criteria in a new table added to the manuscript.
line 109: some additional explanation would be expected as aEEG is applied in most centers that perform HT, as Group C (as I understand from line 159, aEEG was also used in the case of the girl described);
To the best of our knowledge, there is no official "Group C" classification. aEEG is optional and not mandatory for qualification. In our center, aEEG is routinely performed and was used in our patient’s case, although it is not required to initiate HT.
lines 123 - 125: I miss more specific recommendation on timing of MRI; in most centres it is performed between day 5 and 7; according to line 191, in your case it has been performed after day 26? - so it could hardly be characterised as "initial" (line 248);
Thank you for pointing this out. The MRI was performed only on day 26 due to the critical condition of the patient, which made earlier imaging unfeasible. The word “initial” was therefore inappropriate and has been removed.
line 159: "aEEG recording after the initial 10 minutes" - unclear, please specify: after birth? / after admission to NICU? - after initiation of HT?;
The aEEG was recorded 10 minutes after admission to the intensive care unit and before the initiation of HT. This clarification has been added to the text.
line 168: more data on "high pressures" are needed (in cm H2O);
We have included the relevant ventilator settings: PEEP 8 cm H₂O, PS 18 cm H₂O.
line 173: words in brackets regarding active component furosemide are abundant;
We have removed the reference to the active substance from the text.
lines 174 - 177: I doubt that furosemide was the cause for high urea / creatinine (I also miss the data on timing of these high values), more likely it was the result of systemic perinatal asphyxia;
We have rephrased this part to ensure it does not suggest furosemide was the cause of elevated urea/creatinine levels.
line 178: US is not the method to confirm brain death;
We fully agree and appreciate the correction. We have edited the text so as not to imply that ultrasound can confirm brain death.
line 194: as "vision" is not tested, I would suggest to use expressions "an ophtalmologic examination revealed no abnormalities";
This sentence has been revised as suggested.
line 197: how long (in days) did the infant need tube feeding?;
The infant required nasogastric tube feeding until day 41 of life. This information has been added to the manuscript.
The article contains a lot of listing of signs/symptoms/assessments without a "red" thread or structured description (e.g. lines 180 - 185).
We have implemented multiple major and minor edits throughout the manuscript to improve structure and coherence, following your guidance.
A few typo mistakes:
line 51: 0. 98 (correct: 0.98) per 1,000 live births; line 100: pH of 7. 0 (correct 7.0); line 215: Rehabilitatio0n.
Thank you for spotting these errors. They have all been corrected.
Additionally, we have carefully reviewed the entire manuscript and implemented necessary corrections. We have also prepared supplementary materials that will be included as appendices. However, incorporating them directly into the main text would have disrupted the readability of the manuscript.
Best regards,
Marcelina Powązka
